# A Multifocal Study Investigation of Pyrolyzed Printed Circuit Board Leaching



Gvozden Jovanović [1], Mladen Bugarčić [1], Nela Petronijević [1], Srecko R. Stopic [2], Bernd Friedrich [2], Branislav Marković [1,*], Srđan Stanković [3] and Miroslav Sokić [1]

1 Institute for Technology of Nuclear and Other Mineral Raw Materials, 11000 Belgrade, Serbia
2 IME Process Metallurgy and Metal Recycling, RWTH Aachen University, 52056 Aachen, Germany
3 Federal Institute for Geosciences and Natural Resources, Stilleweg 2, 30655 Hannover, Germany
* Correspondence: b.markovic@itnms.ac.rs; Tel.: +381-11-3691-722

**Abstract:** Electric waste from numerous devices that are put out of use every day has some form of printed circuit board that contains precious and valuable metals in their components. In order to extract these metals, the printed circuit boards were crushed and pyrolyzed into powder. The fine pyrolyzed printed circuit board (PPCB) powder was separated into fractions, and the fine metallic fraction was used as a raw material for metal leaching extraction. In order to better understand how various metal species react in leaching media, several leaching agents were used (sulfuric acid, nitric acid, glycine, and acid mine drainage-AMD) alone, and with the addition of hydrogen peroxide. Additionally, the influence of the S/L ratio and leaching temperature were investigated in sulfuric acid leaching solutions, as this is the one most widely used. In one case, the reactor was heated in a thermal bath, while in the other, it was heated in an ultrasonic bath. Lastly, several experiments were conducted with a (consecutive) two-pronged leaching approach, with and without applied pretreatment. The aim of this paper is to give a multifocal and detailed approach to how metals such as Al, Cu, Co, Zn, Sn, and Ca behave when extracted from fine PPCB powder. However, some attention is given to Nd, Pd, Pb, and Ba as well. One of the main findings is that regardless of the pretreatment or the sequence of leaching media applied, consecutive two-pronged leaching cannot be used for selective metal extraction. However, AMD was found to be suitable for selective leaching with very limited applications.

**Keywords:** e-waste; hydrometallurgy; acidic media; oxidizing agent; metal extraction

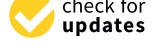



## 1. Introduction

Electronic devices have become commonplace in modern society's daily life. The global market provides numerous devices that we commonly use, such as smart phones, tablets, laptops, and personal computers. All of these devices are made out of different metals required for construction of printed circuit boards (PCBs), which are an essential functional component of any electronic device. As electronics consumption increases, so does the amount of electronic waste (e-waste). According to the most recent Global E-waste Monitor report from 2020 (GEM 2020), 53.6 million metric tons of e-waste were generated in 2019 and an upward trend is anticipated [1]. E-waste, if not further processed as a secondary resource or safely disposed of, may cause serious environmental damage because it is rich in heavy metals and metalloids, both of which are toxic to the biosphere. The element distribution in the PCBs varies due to the application of the devices for which they are designed. However, in its highest content it consists of metals and metalloids (as low as ≈ 21 [2] up to ≈ 42 wt% [3]), ceramics (in a range from 15 [4] to 59.26 wt% [5]), and various polymer materials (in a range from 10 to 30 wt% [6]), illustrated in Figure 1.

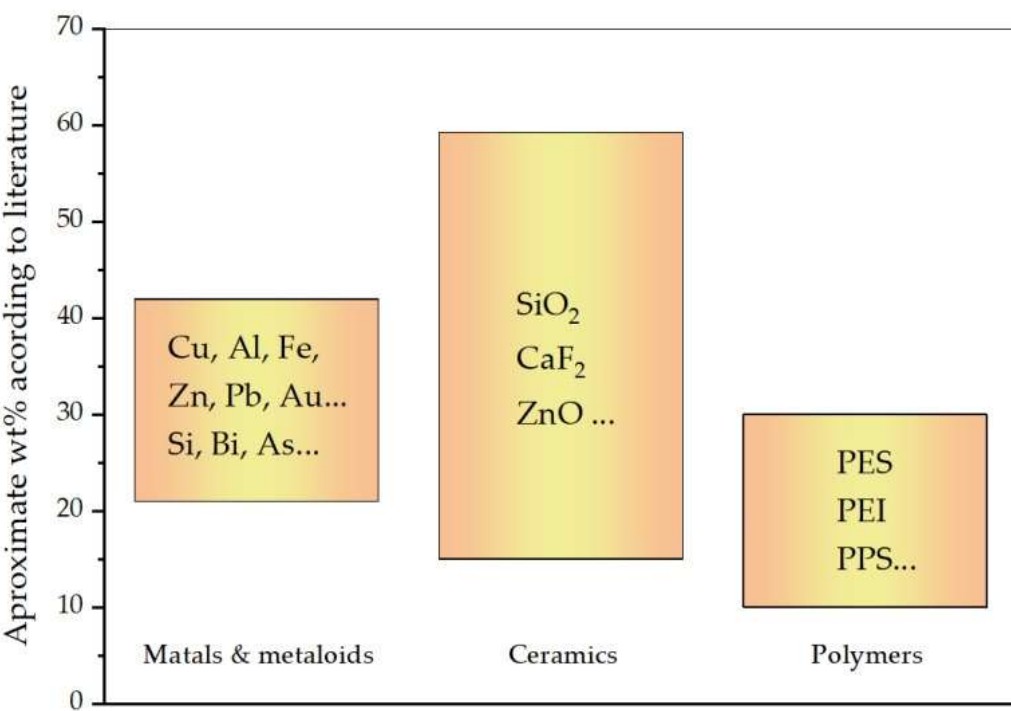

**Figure 1.** Range of printed circuit board constituents with their possible material content.

Metals and metalloids present in PCBs are the most valuable resource since the amount of gold, platinum, palladium, silver, and other non-precious but expensive metals are much higher than the richest ores [7].

Mechanical dismantling, which is dismantling via heat treatment and chemical methods separation, can be used to separate electronic components from the waste PCBs [8]. All the mentioned procedures in the work of Maurice et al. [8] are used on the industrial scale with all of their benefits, drawbacks, costs, and simplicities detailed, some are even patented technologies. However, hydrometallurgy procedures are the least hazardous to the environment and they are very simple to perform. The literature preview gives additional assets to hydrometallurgy over pyrometallurgy procedures, such as slag formation [9] and low quality of the product due to complications that can arise during the pyrolysis process [10]. Hydrometallurgy, compared with pyrometallurgy, plants have lower production capacities, and consequently significantly lower investment requirements, lower energy consumption, reduced emissions, higher predictability and, finally, a higher degree of flexibility and easier control of the production process [11]. Mechanical pretreatment is another method that contributes to materials separation because PCBs are made up of both magnetic and paramagnetic materials, those with higher and lower density, and combustible and non-combustible materials. Keeping all of this in mind, mechanical pretreatment would undoubtedly improve the separation process and should not be avoided [12].

To achieve a high recovery rate of the targeted metal present in e-waste, it is necessary not only to liberate the surface of the metal to the leaching medium, but also to modify those materials that hinder the reaction with leaching substances, such as polymers [13]. One of the methods suitable for polymer deformation and degradation is oxidative pyrolysis, which is suitable not only for polymer degradation but also for partially oxidizing the surface of the metals present in PCBs such as copper, zinc, tin, aluminum, and others that can readily oxidize.

A review of the literature provides several examples of leaching mediums that could be used to maximize the extraction of valuable materials from waste PCB, among them: mineral acids (sulfuric [14,15], nitric [16,17], and hydrochloric [18,19]), together with suitable

oxidizing agents such as hydrogen peroxide [15,20,21] and halogens [22,23]. In addition to acidic leaching mediums, strong bases such as sodium hydroxide [24], potassium hydroxide [25], or their mixture [26] could be utilized. In order to leach noble metals from PCBs, several lixiviants may be used, such as cyanides [27], thiosulfates [28], thiourea [29], ammonia [30], and persulfate [31] but also a recent focus on deep eutectic solvents (DES) should not be neglected.

Some of the most utilized DES leaching mediums are those based on choline chloride (ChCl), such as reline ($X_{ChCl/urea}$ = 1:2), ethaline ($X_{ChCl/ethylene\ glycol}$ = 1:2), or its mixture with malonic acid ($X_{ChCl/malonic\ acid}$ = 1:1) [32]. In addition to those mixtures, there are also a group of natural deep eutectic solvents (NADES), most of which are $\alpha$-amino acids which may be used both as a pair of hydrogen bond donor and acceptor (HBD and HBA) [33] but also alone for their chemical properties that include acting as bidentate ligands. Among all of $\alpha$-amino acids, the highest interest has arisen towards glycine, since it is easily accessible as a resource [34], but even more thanks to its leaching potential in terms of leaching rate [35]. Glycine is nontoxic, biodegradable, and an easily accessible lixiviant; therefore, it can be considered as one of the candidates for optimal metal leaching from waste PCBs.

Recovering precious metals from pyrolyzed printed circuit board (PPCB) powder is a reasonable option, since leaching fine PPCB powder is much less toxic than PCB residue [36]. Several commonly used leaching mediums such as sulfuric acid, nitric acid, sodium hydroxide, and glycine were compared with regard to leaching metals from fine PPCB powder. They were tested individually, consecutively, and with or without common oxidizing agents such as hydrogen peroxide and air. Furthermore, acid mine drainage (AMD) was also tested to see if it can be a concurrent leaching medium. According to the best of our knowledge, there are no studies with such a broad scope of investigations in the literature.

## 2. Materials and Methods

Printed circuit boards from smart phones were pyrolyzed in a nitrogen atmosphere for 90 min at a heating rate of 300 °C/h at 570 °C. The starting mass of the PCBs was 5535 g and after pyrolysis and grinding it was 4088 g. This mass loss of 24.8% is contributed to the pyrolysis transition of polymer plastics to carbon. This mass of PPCB can be separated into fractions above 500 μm (1612 g) and below 500 μm (2476 g). The larger fraction mainly consisted of carbon, glass, and metals, while the smaller fraction can further be magnetically separated into non-metallic (909 g) and metallic (1451 g) fractions. The fine metallic fraction (1451 g) was the target of our leaching investigations and is labeled as fine PPCB powder and its elemental distribution is given in Table 1. This was conducted by Inductively Coupled Plasma Optical Emission Spectroscopy (ICP-OES) analysis based on chemical digestion (Spectro Arcos, SPECTRO Analytical Instruments GmbH, Kleve, Germany). All digestions led to full dissolution of samples. Chemical digestions of fine PPCB powder were performed with concentrated acids at high temperatures and high pressures. Sample quantities amounted between 250 mg and 800 mg per digestion.

**Table 1.** Element distribution of fine PPCB powder.

| Element * Concentration (mg/g) | | | | | | | | | | |
|---|---|---|---|---|---|---|---|---|---|---|
| Ag | Au | Al | Ba | C | Ca | Ce | Co | Cu | Fe | Li |
| 0.180 | 0.180 | 48.1 | 8.30 | 177 | 62.4 | 0.350 | 0.042 | 115 | 11.3 | 0.023 |
| Nd | Ni | Mn | Pb | Pd | Pt | Si | Sn | Y | Zn | |
| 0.218 | 1.9 | 0.623 | 14.3 | 0.203 | 0.044 | 100 | 26.0 | 0.013 | 9.00 | |

\* Origin of each element is described in Supplementary Materials.

The AMD sample was collected from Mpumalanga, South Africa. All sampling and laboratory analysis was performed in accordance with recognized global standards such as

the International Standards Organization (ISO), as shown in Table 2. The pH of the AMD wastewater was around 2.0.

**Table 2.** Element distribution of acid mine drainage (AMD).

| Element Concentration (mg/L) | | | | | | | | | | |
|---|---|---|---|---|---|---|---|---|---|---|
| Ag | Au | Al | Ba | Mg | Ca | Ce | Co | Cu | Fe | Li |
| <0.5 | <0.5 | 423 | <0.5 | 428 | 491 | 4.91 | 1.22 | 1750 | 2630 | <0.5 |
| Nd | Ni | Mn | Pb | Pd | Pt | Si | Sn | Y | Zn | La |
| 1.1 | 1.26 | 8.88 | 0.76 | <0.5 | <0.5 | 44.9 | <0.5 | 1.54 | 8.61 | <0.5 |

### 2.1. Experimental Setup

During our experiments, two types of reactor setups were used: closed and open. Figure 2a depicts the closed experimental set-up, which includes a reactor, stirrer, Allihn condenser, and a thermal bath. Figure 2b depicts the open experimental set-up, which includes a reactor, stirrer, ultrasonic bath, and an air compressor, as well as a rotameter and a gas washing bottle.

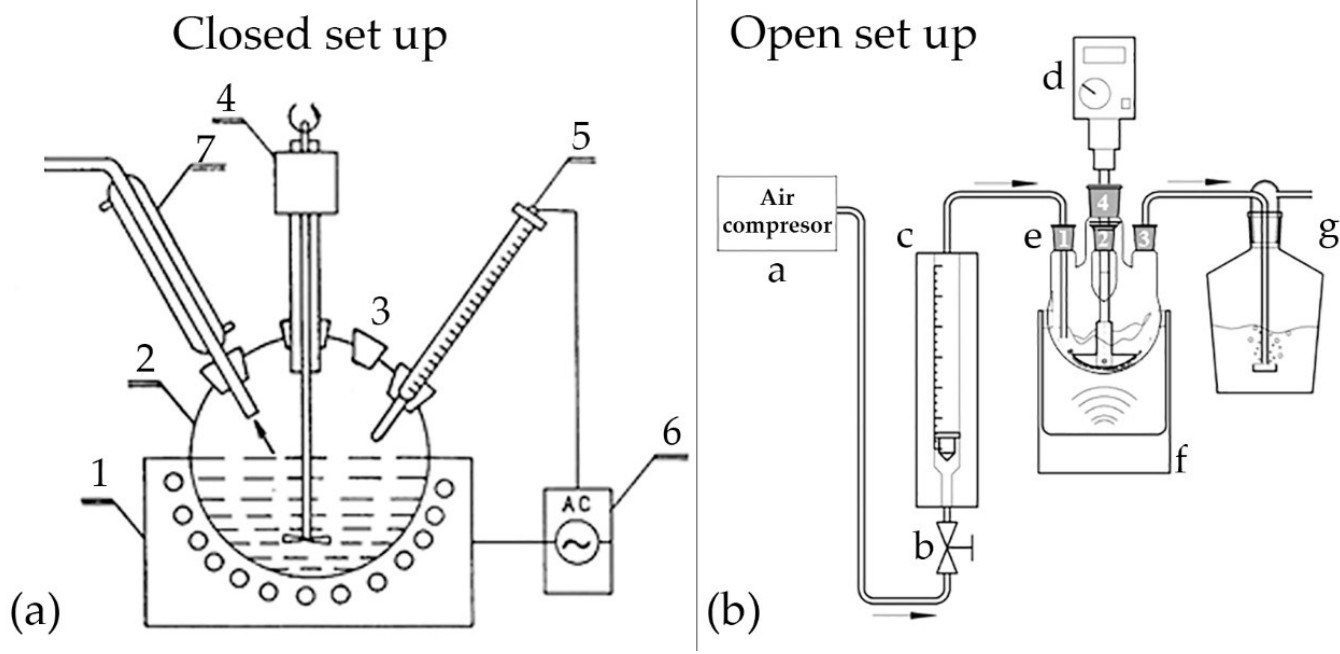

**Figure 2.** Experimental set-up: (**a**) closed set-up—1 (thermal bath), 2 (reactor), 3 (sampling neck), 4 (mechanical stirrer), 5 (thermocouple), 6 (thermoregulation), 7 (Allihn condenser); (**b**) open set-up—a (air compressor), b (air flow valve), c (rotameter), d (mechanical stirrer), e (reactor), f (ultrasonic bath), g (gas washing bottle), 1 (air injecting neck), 2 (sampling neck), 3 (air outlet neck), 4 (stirrer seal neck).

As shown in Figure 2, in the closed set-up, consisted of "PT100" thermocouple (Temperatur Messelemente TMH, Hettstedt GmbH, Maintal, Germany), thermal bath with thermoregulation (SAF Wärmetechnik GmbH, Weinheimer Str. 2A, 69509 Mörlenbach, Germany), and the stirrer unit consisted of a motor unit with a drill chuck, type "IKA Eurostar digital" (IKA®-Werke GmbH & Co.KG, Staufen, Germany). The reactor vessel was four-necked round-bottom flasks with a capacity of 2000 mL.

In the open set-up, the stirrer unit consisted of a motor unit with a drill chuck, type "IKA Eurostar digital" (IKA®-Werke GmbH & Co.KG, Staufen, Germany), and a polytetrafluorethylene (PTFE) coated impeller including a PTFE stirrer seal. The ultrasonic bath used was a Bandelin Sonorex RK 52H (BANDELIN electronic GmbH & Co, KG, Berlin, Germany), and the type of rotametar used was Rota Yokogawa (Yokogawa Deutschland

GmbH, Ratingen, Germany). The device used for oxi-reduction potential (ORP) measurements was a pH-meter 7310 (InoLAb, WTW, Weilheim, Germany). The reactor vessel was four-necked round-bottom flasks with a capacity of 500 mL.

For the closed experimental set, three series of experiments were carried out:

- Series 1—Solid/liquid ratio investigations on sulfuric acid leaching (high and low);
- Series 2—Influence of hydrogen peroxide concentration on sulfuric acid leaching;
- Series 3—Acid mine drainage and distilled water leaching with the aid of hydrogen peroxide.

For the open experimental set, two series of experiments were carried out:

- Series 1—Influence of hydrogen peroxide on NADES and nitric acid leaching;
- Series 2—Influence of pretreatments on two-pronged leaching.

### 2.2. Leaching Conditions

For the closed set series 1, for a low S/L ratio, 1200 mL of sulfuric acid in various concentrations (0.5 M, 1 M, and 2 M), and three starting masses of PPCB (30 g, 50 g, and 70 g) were used, with the leaching solution kept at 60 °C. However, one experiment was conducted with 3 M sulfuric acid at a S/L ratio of 0.025 (30 g of PPCB). Furthermore, two experiments were conducted at different temperatures, 40 °C (1 M $H_2SO_4$, 50 g PPCB) and 80 °C (0.5 M $H_2SO_4$, 30 g PPCB).

In all four experiments for the closed set series 1, for a high S/L ratio, 200 mL of 2 M $H_2SO_4$ was used, and the temperature of the leaching solution was kept at 60 °C while the starting mass of PPCB was 20, 30, 40, and 60 g. The low S/L ratio experiments were sampled at 1, 2, and 4 h, while the high S/L ratio experiments were sampled at the end, after 4 h.

For the closed set series 2, 200 mL of 2 M $H_2SO_4$ was used as a leaching agent, and the leaching solution temperature was kept at 60 °C. At the start of the experiment, three different concentrations of hydrogen peroxide were used: 1 M $H_2O_2$ (23.46 mL), 2 M $H_2O_2$ (46.92 mL), and 3 M $H_2O_2$ (70.38 mL). The starting mass of PPCB for the experiments with the 2 M $H_2O_2$ was 20, 30, 40, and 60 g, while the starting mass for the experiments with the 1 M and 3 M $H_2O_2$ was 20 and 40 g, respectively. For all of the experiments in series 2, samples were collected after 4 h.

In the closed set series 3, 200 mL of AMD was used as a leaching agent on 20 g of PPCB in the AMD experiments, and the temperature of the leaching solution was kept at 60 °C. One experiment was conducted only with AMD, and three others were conducted with different concentrations of hydrogen peroxide: 1 M $H_2O_2$ (23.46 mL), 2 M $H_2O_2$ (46.92 mL), and 3 M $H_2O_2$ (70.38 mL). All four experiments were repeated with the addition of 5 mL of concentrated sulfuric acid. Furthermore, for the water experiments, 200 mL of distilled $H_2O$ was used as a leaching agent on 20 g of PPCB, with the leaching solution kept at 60 °C. The two experiments were carried out with the addition of 1 M $H_2O_2$ (23.46 mL), and 3 M $H_2O_2$ (70.38 mL). For all of the experiments in Series 3, samples were collected after 4 h.

The ultrasound sonification was turned on for the open set regardless of the series, and the ultrasound bath was heated to 60 °C. The initial solid–liquid ratio used was 0.1 (15 g of PPCB in 150 mL of leaching solution), and the airflow injected into the reactor ranged between 2 and 2.5 L/min. The stirring speed was around 240 rpm. The leaching time of the experiments was 8 h, with samples taken every 2 h, and the ORP was measured. Due to the open nature of the reactor setup, 16 mL of distilled water was added hourly (15 min before sampling) in order to keep the volume of the leaching solution as constant as possible. The 8 mL of distilled water (out of 16 mL) was replaced by 3 M $H_2O_2$ in the experiments that added hydrogen peroxide.

There are three leaching experiments in open set series 1. The first used nitric acid as the leaching agent at a concentration of 2 mol/L, with hydrogen peroxide added on a regular basis. The other two used NADES, which is a 2 mol/L solution of glycine ($NH_2$-$CH_2$-$COOH$) that was adjusted to a pH value of around 12. This is accomplished by adding

small amounts of 2 M NaOH. One NADES experiment included the addition of hydrogen peroxide, while the other did not.

Three two-pronged leaching experiments were carried out for open set series 2. Two-pronged leaching means that one leaching agent was used for the first 4 h, then the leaching residue was filtered, rinsed, and dried, and the leaching was repeated for another 4 h with a different leaching medium. The leaching solution was kept at 150 mL for both leaching steps. The first leaching agent used in the first experiment was 1 M sulfuric acid, and the second leaching agent used was 2 M nitric acid, with no pretreatment employed. For the other two experiments the first leaching agent used was 2 M nitric acid, and the second leaching agent was NADES with added hydrogen peroxide, with two different pretreatments employed. One experiment had a "chemical pretreatment", which consisted of an 8 h leaching in 2 M NaOH at 70 °C with ultrasound but no airflow (closed system). The other experiment had a "physical pretreatment", in which swelling of the pyrolyzed polymers was attempted by adding trichloroethylene ($C_2HCl_3$) to the fine PPCB powder mass, mixing it in a mortar and pestle, and drying it until constant mass.

## 3. Results and Discussion

The leaching degree of various metallic species extracted by leaching agents such as sulfuric and nitric acids as well as NADES (glycine with NaOH) is calculated by the following Equation:

$$\text{Leaching degree (\%)} = \frac{C_x \cdot V_x}{C_0 \cdot m_0} \times 100\% \tag{1}$$

where $C_x$ is the metal concentration in the leachate after a certain number of hours, $V_x$ is the volume of the leachate solution, $C_0$ is the metal concentration in the starting fine PPCB powder, and $m_0$ is the mass of the fine PPCB powder used for leaching.

However, if the primary leaching agent used is acid mine drainage (AMD), the elemental distribution of which is given in Table 2, regardless of whether the aid of concentrated sulfuric acid or hydrogen peroxide was used, then the leaching degree is calculated differently:

$$\text{Leeching degree (\%)} = \frac{C_x \cdot V_x - C_a \cdot V_a}{C_0 \cdot m_0} \times 100\% \tag{2}$$

where $C_a$ is the metal concentration in the AMD and $V_a$ is the volume of the AMD used for leaching, while the rest is the same as in Equation (1).

### 3.1. Solid/Liquid Ratio Investigations on Sulfuric Acid Leaching

Figure 3 depicts the overall effect of the starting S/L ratio, where all metals except for Co, Zn, and Sn showed similar, descending trends of leaching degree with increasing S/L ratio. This is expected because the higher the S/L ratio, the less leachable media there is to extract the metals into, increasing the insufficiency of the acid and decreasing the amount of metal leached. This happens due to mass transfer limitations, which occur when the leaching rate is limited by the diffusion of metals from the solid surface to the bulk solution. It can be concluded that the leaching reactions for Al, Cu, and Ca are diffusion limited, with almost linear trendlines from 0.1 to 0.3 g mL$^{-1}$.

The presence of cobalt in the PPCB can be contributed to lithium cobalt oxide ($LiCoO_2$) that is readily used as a cathode in lithium-ion batteries. Therefore, lithium is present as ion exchangeable cation in the structure of $LiCoO_2$, and the rate of extraction is proportional to the ion exchange mechanism, as explained by reaction (3). The penetration of hydrogen sulfate anion through the interlayer leaches lithium as $LiHSO_4$, where the imbalanced positive charge is compensated with hydronium cations ($H_3O^+$). This mechanism is detrimental to the leaching of cobalt because the octahedrons consist of an octahedral cage whose edges are $O^{2-}$ ions and whose center is $Co^{3+}$ ions. For this reason, cobalt cations are not easily leached with bisulfate anions in the absence of other metals that could be oxidized. As a result, leaching is disrupted due to an insufficiency of metal (M) oxidized by $Co^{3+}$ ions when the S/L ratio is low. After reaching a maximum at a S/L ratio = 0.1 g mL$^{-1}$,

the leaching degree no longer increases with a further increase in the S/L ratio, because leaching is now limited by the diffusion rate.

$$2LiCoO_{2(s)} + 4H_2SO_{4(aq)} + M \rightarrow 2LiHSO_{4(aq)} + 2CoSO_{4(aq)} + MSO_{4(aq)} + 3H_2O_{(l)} \quad (3)$$

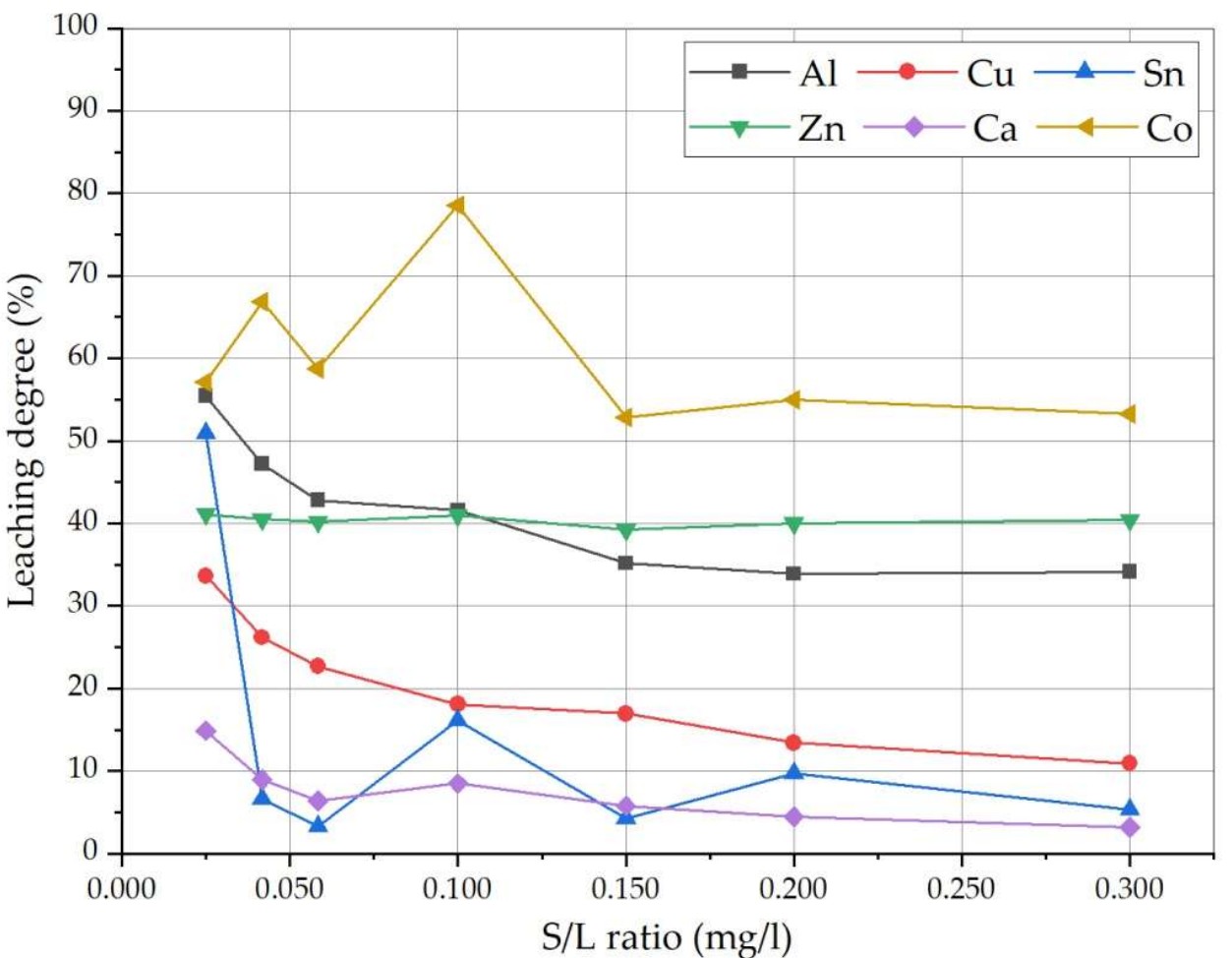

**Figure 3.** Influence of starting solid liquid ration on 2 M sulfuric acid leaching after 4 h.

Calcium fluoride is well known for its insulating or semiconducting electrical properties; therefore, the presence of calcium in PPCB can be contributed to $CaF_2$ [37]. Furthermore, calcium fluoride is poorly soluble in water, with a solubility product of $4.0 \times 10^{-11}$. The ion exchange reaction (4) represents how calcium fluoride behaves in sulfuric acid. When this reaction is in equilibrium, the calcium sulfate backward reaction is dominant since calcium sulfate's solubility in water is slightly higher than that of calcium fluoride ($K_{sp} = 3.7 \times 10^{-5}$).

$$CaF_{2(s)} + H_2SO_{4(aq)} \leftrightarrows CaSO_{4(s)} + 2HF_{(aq)} \quad (4)$$

However, the chemical equilibrium in reaction (4) is disrupted by reaction (5), in which the hydrofluoric acid produced by $CaF_2$ leaching reacts with $SiO_2$. Silica is abundant due to the presence of small glass fibers in many semiconductors. This changes the reaction (4) from a backward to a forward dominant reaction. Despite the involvement of reaction (5), the degree of calcium fluoride leaching is very limited because reaction (4) produces a small amount of hydrofluoric acid that can react with PPCB silica components.

$$SiO_{2(s)} + 6HF_{(aq)} \longrightarrow H_2SiF_{6(aq)} + 2H_2O_{(l)} \quad (5)$$

The two-component alloy of lead and tin, Sn63/Pb37, is used as an effective solder material. Although the use of lead is minimized since it has a high toxicity as a heavy metal, solder with this composition is still present in PCBs. Sn can occur in three oxidation states: 0, +2, and +4. During the hydrometallurgical treatment, the 0 (Sn) and +4 ($SnO_2 \cdot xH_2O$) compounds are insoluble while the +2 ($SnSO_4$) compound is a salt; readily soluble in water. The leaching of tin in sulfuric acid for the formation of $SnO_2 \cdot xH_2O$ follows reaction (7), whereas reaction (6) is beneficial and results in the calculated leaching degree.

$$Sn_{(s)} + 2H_2SO_{4(aq)} \rightarrow SnSO_{4(aq)} + SO_{2(g)} + 2H_2O_{(l)} \tag{6}$$

$$3Sn_{(s)} + 2H_2SO_{4(aq)} + 2O_{2(g)} + 4H_2O_{(l)} \rightarrow 3Sn(OH)_{4(s)} + 2SO_{2(g)} \tag{7}$$

The amount of dissolved oxygen present in the leaching solution is the main factor that determines which reaction will be more prevalent, because an increase in oxygen concentration is directly beneficial for reaction (7). The impact of mass transfer in the leaching solution not only aids the oxidation process but also raises the reaction rate in both cases, increasing the overall leaching degree at low S/L ratios.

The presence of aluminum in PCBs can be attributed to both its elemental metallic form and various semiconducting or electrically resistant ceramics. One of these ceramics is $CeMgAl_{11}O_{19}$, with Al present in low amounts because it comes from a luminescent source support matrix [38]. The major portion of aluminum in elemental form comes from heat exchangers for various electronic devices, the use of which generates heat; thus, reactions that focus on elemental aluminum would give the most adequate explanation as to its trends in leaching degree.

Aluminum, as is well known, easily forms passive oxide film during its reaction with aggressive corrosion agents, as shown in Equations (8) to (10).

$$2Al_{(s)} + 3H_2SO_{4(aq)} \rightarrow Al_2(SO_4)_{3(aq)} + 3H_{2(g)} \tag{8}$$

$$2Al_{(s)} + 6HNO_{3(aq)} \rightarrow 2Al(NO_3)_{3(aq)} + 3H_{2(g)} \tag{9}$$

$$2Al_{(s)} + 2NaOH_{(aq)} + O_{2(g)} \rightarrow 2NaAlO_{2(aq)} + H_{2(g)} \tag{10}$$

As a result, the corrosion approach would be the most logical way to explain Al leaching. Furthermore, it should be noted that the degree of Al leaching would theoretically be the highest at the temperature of our investigation (60 °C), because the cathodic reaction of oxygen reduction (11) reaches the maximum current density and uses the most oxygen from the leaching solution [39]. This decrease in oxygen concentration in the solution causes the thinning of the passive oxide film, allowing for further leaching of Al.

$$O_2 + 4H^+ + 4e^- \rightarrow 2H_2O \tag{11}$$

As stated above, with the increase in the S/L ratio, and the changes that this increase creates in mass transfer, diffusion becomes the limiting factor that contributes to the leaching degree of Al, not only of the metal to the leaching solution but of oxygen as well.

It goes without saying that the presence of copper is unavoidable in any electronic device, this is especially true for PCBs. The leaching of copper by sulfuric acid (12) in the presence of dissolved oxygen happens spontaneously, and this is the main reaction of copper oxidation (12). Another reaction that is involved in copper oxidation is the reaction with readily present iron (III) species, where iron (III) etches the copper surface, resulting in copper oxidation (13) [40].

$$2Cu_{(s)} + 2H_2SO_{4(aq)} + O_{2(g)} \rightarrow 2CuSO_{4(aq)} + 2H_2O_{(l)} \tag{12}$$

$$Cu_{(s)} + 2Fe^{3+}_{(aq)} \rightarrow Cu^{2+}_{(aq)} + 2Fe^{2+}_{(aq)} \tag{13}$$

Although copper does not form a thick passive layer like aluminum, its leaching rates follow a similar pattern, not only as the S/L ratio increases, but also as sulfuric acid concentrations and leaching temperature change [41].

Unlike Al and Cu, zinc is not present in its elemental metallic form, but in forms of zinc oxide and zinc sulfide that are used as semiconducting components in electronic devices. The leaching of zinc sulfide requires the aid of dissolved oxygen, while zinc oxide does not, as can be seen in Equations (14) and (15). Furthermore, as research of zinc leaching from sphalerite by any acid media showed, elemental sulfur [42,43], as well as other insoluble salts (lead (II) sulfate and barium sulfate), can form on the surface of zinc sulfide hindering any further leaching [44,45].

$$ZnO_{(s)} + H_2SO_{4(aq)} \rightarrow ZnSO_{4(aq)} + H_2O_{(l)} \tag{14}$$

$$2ZnS_{(s)} + 2H_2SO_{4(aq)} + O_{2(g)} \rightarrow 2ZnSO_{4(aq)} + S_{(s)} + 2H_2O_{(l)} \tag{15}$$

Due to the nature of this hindering mechanism, the leaching of zinc is least affected by changes in the S/L ratio, acid concentration or leaching temperature.

### 3.2. Acid Concentration Effect on PPCB Leaching

For the study of the influence of sulfuric acid concentrations, four different concentrations were investigated (0.5 M, 1 M, 2 M, and 3 M), all of them for the smallest S/L ratio 0.025, and two for 0.058 g mL$^{-1}$.

Figures 4 and 5 show how the leaching degrees (Al, Cu, Sn, and Zn) change depending on how long the fine PPCB powder is in contact with different concentrations of leaching solutions. These findings can be combined with the effect of the S/L ratio for the four hours leaching in 2 M sulfuric acid. They show how Al, Cu, Sn, and Zn behave during leaching.

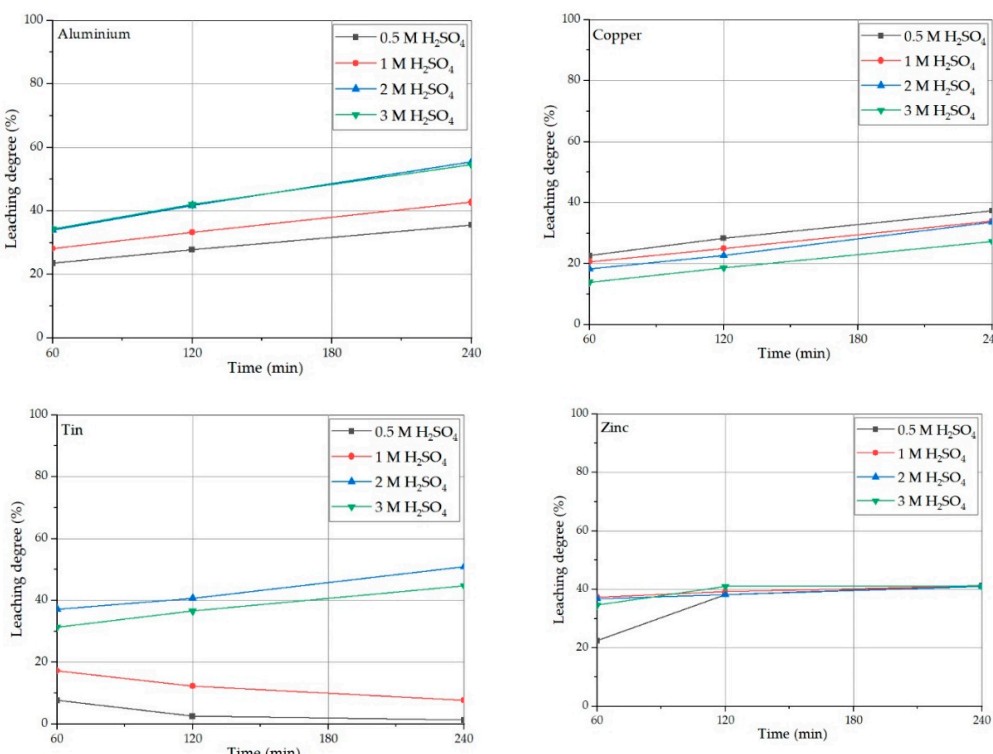

**Figure 4.** Influence of sulfuric acid concentration on leaching for a S/L ratio of 0.025 (30 g of PPCB).

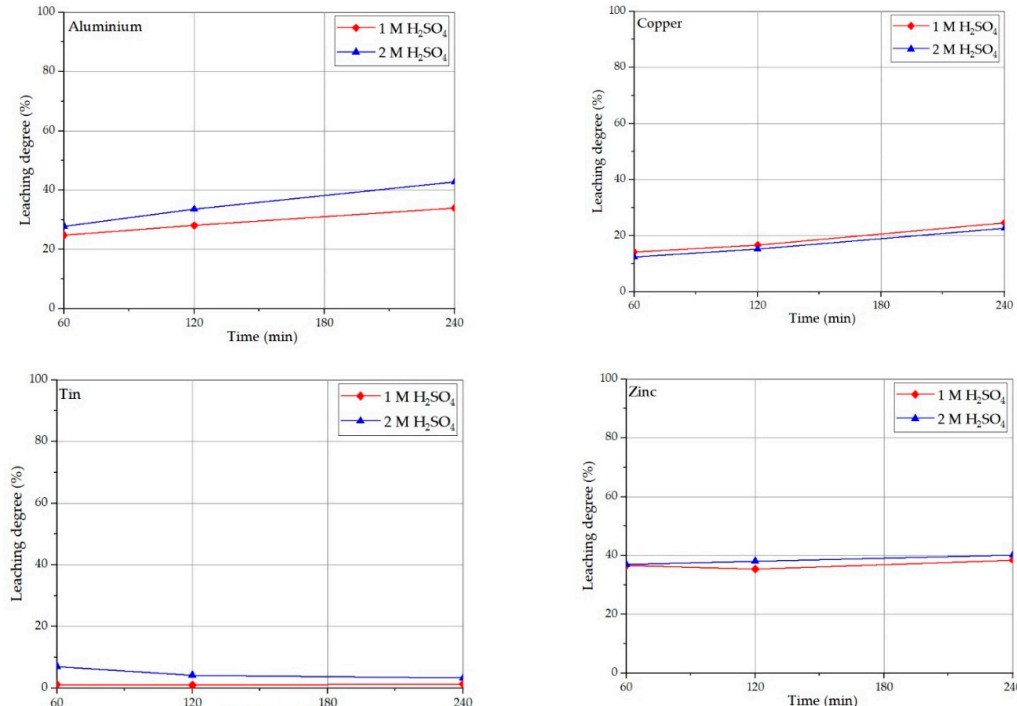

**Figure 5.** Influence of sulfuric acid concentration on leaching for a S/L ratio of 0.058 (70 g of PPCB).

According to the corrosion approach, aluminum leaching degrees overlap in 2 M and 3 M sulfuric acid (Figure 4) because acid concentration lowers oxygen solubility, which thins the passive oxide film and promotes Al recovery. Sulfuric acid concentration increases the initial rate of aluminum corrosion [46]. As the S/L ratio increases, acid concentration has little effect on aluminum leaching (Figures 4 and 5).

As previously stated, as sulfuric acid concentrations rise, the oxygen solubility decreases, affecting copper leaching in a different way despite the fact that copper does not form a passive layer like aluminum. Lowering the available oxygen in the leaching solution severely disrupts the formation of copper (II) sulfate as shown in Equation (13), explaining the decrease in leaching degree seen in Figures 4 and 5.

According to Equation (6), tin leaching is better without oxygen, because oxygen produces insoluble $SnO_2 \cdot xH_2O$. Dissolved oxygen is less disrupted in 0.5 M and 1.0 M sulfuric acid. Leaching that produces $SnO_2 \cdot xH_2O$ and $SnSO_4$ are concurrent, thus lowering the total leaching degree. Figures 4 and 5 show the difference in sulfuric acid leaching caused by this effect. This suggests that Sn (II) oxidizes to Sn (IV) at a critical acid concentration of between 1 M and 2 M for the low S/L ratio of 0.025. Mass transfer of reaction participants, mainly oxygen and Sn (II), prevents this phenomenon at 0.058 S/L ratio.

Regardless of the S/L ratio, the behavior of the zinc leaching degree with increasing acid concentration is very peculiar. For all acid concentrations above 0.5 M, the effectiveness seems to reach a plateau around 40%, sometime before the first hour. This is most likely due to all of the zinc oxide being expended, leaving only zinc sulfide, which passivates and easily stops the leaching process. This occurs by the second hour for the lowest concentration of sulfuric acid.

### 3.3. Temperature Influence on Leaching Effectivity

Two sulfuric acid concentrations, 0.5 M and 1 M, were used to study how temperature affects Zn, Ca, Al, and Cu leaching. Leaching with 0.5 M acid was conducted on 30 g fine PPCB powder (S/L = 0.025) at 60 and 80 °C, while leaching with 1 M acid was conducted on 50 g (S/L = 0.042) at 40 and 60 °C.

Just as with the acid concentration investigation, the leaching degree of zinc reaches a plateau around 40%. There is a slight difference, however, with a change in leaching

temperature. For the temperature of 80 °C, the plateau is around 45% while for 40 °C, the plateau is around 35%. For the lower acid concentration of 0.5 M the plateau is still achieved by the second hour, but for 80 °C the plateau is reached at a higher rate. Since calcium sulphate and its crystal hydrates are poorly soluble in water, all calcium leaching rate trendlines in Figure 6 plateaued after 2 h. Calcium leaching depends on sulfate anion concentration ($K_{sp}(CaSO_4, 25\ °C) = 3.7 \times 10^{-5}$ [47]). Red trendlines show expected leaching degrees for the same concentration of acid but at a different temperature (higher temperature–lower solubility [48]), but this trend is not seen for lower concentrations, likely due to gypsum precipitation before quantitative analysis of the solutions. Aluminum corrosion increases with temperature. According to Chen et al. [39], aluminum's corrosion mechanism changes, and oxygen solubility decreases with temperature, lowering the cathodic current and affecting leaching. Concentration of sulfuric acid should increase copper oxidation, but it also lowers oxygen solubility, so copper extraction is preferable at lower concentrations (Figure 6), especially at higher temperatures when the oxygen diffusion coefficient is higher [49].

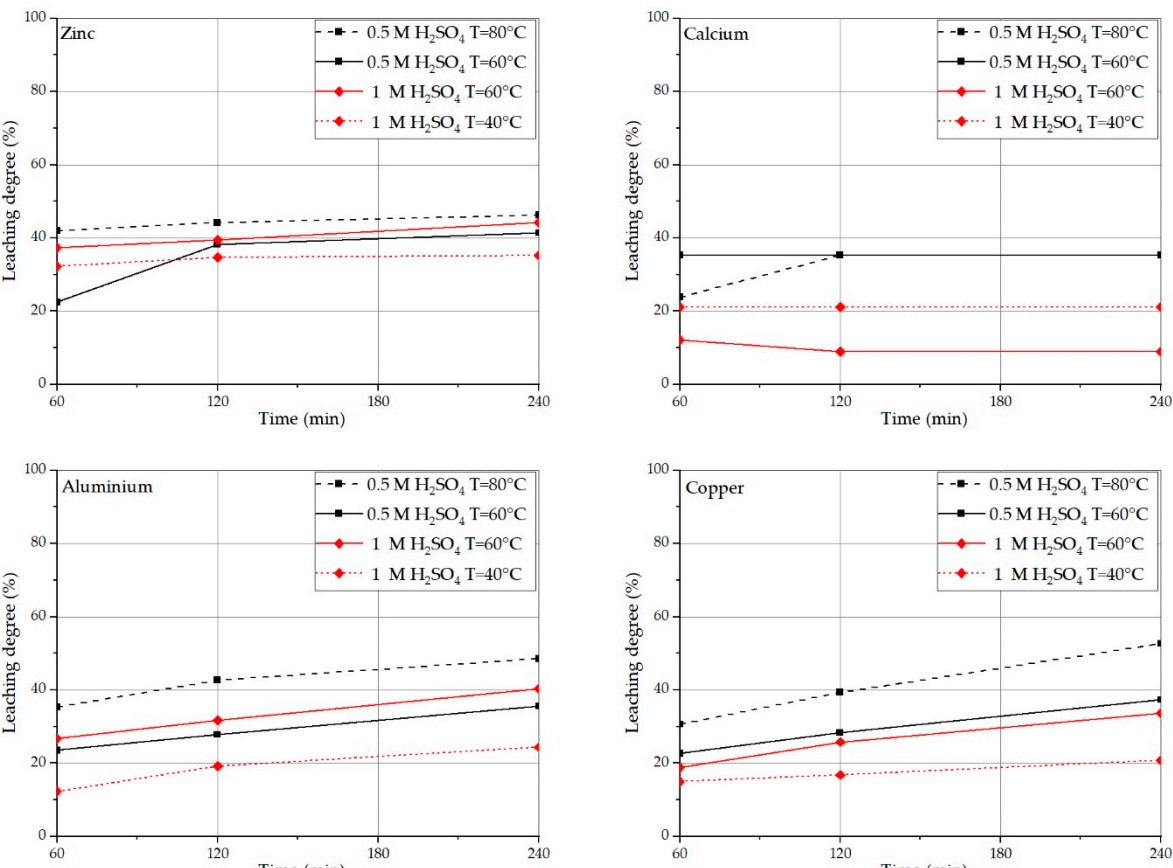

**Figure 6.** Influence of temperature on leaching degree, 30 g PPCB in 0.5 M and 50 g PPCB in 1 M sulfuric acid.

### 3.4. Influence of Hydrogen Peroxide Concentration on Sulfuric Acid Leaching

Hydrogen peroxide is used in the leaching process to replace the dissolved oxygen in the leaching solution. As a result of the higher oxidation potential of hydrogen peroxide compared to dissolved oxygen ($Er(H_2O_2 + H^+/H_2O) = 1.776$ V [47]), it alters most reactions in which oxygen participates, regardless of whether those reaction are beneficial for leaching or not. Furthermore, the concentration of hydrogen peroxide in the leaching solution can be several times higher than the concentration of dissolved oxygen, which is very limited. Just as there are two types of Sn leaching reactions that occur concurrently when no hydrogen peroxide is present (beneficial reaction 6 and detrimental reaction 7), the addition

of hydrogen peroxide to the leaching solution also produces two types of reaction, but one is dominantly prevalent. Reaction (16) shows the beneficial influence of hydrogen peroxide on Sn leaching, which produces $SnSO_4$, while reaction (17) shows the detrimental predominant influence of hydrogen peroxide, which produces $SnO_2 \cdot xH_2O$. Figure 7 shows that the additional amount of hydrogen peroxide has a negative impact on the degree of Sn leaching. Even without hydrogen peroxide, the leaching degree is lower (below 20%) because the S/L ratio is in the 0.1 to 0.3 range.

$$Sn_{(s)} + H_2SO_{4(aq)} + H_2O_{2(aq)} \rightarrow SnSO_{4(s)} + 2H_2O_{(l)} \quad (16)$$

$$Sn_{(s)} + H_2SO_{4(aq)} + H_2O_{2(aq)} \rightarrow Sn(OH)_{4(s)} + SO_{2(g)} \quad (17)$$

One of the characteristics of hydrogen peroxide is its spontaneous decomposition over time, which is depicted in reaction (18). This decomposition generates oxygen bubbles, which eventually escape from the leaching solution. The bubbles aid in the removal of gaseous hydrogen fluoride from the leaching solution. This gaseous hydrogen fluoride is formed from dissolved hydrogen fluoride produced by reaction (4), and decreasing its concentration pushes the equilibrium to a hydrogen–forward direction, directly increasing the degree of calcium leaching. The amount of bubbles formed is proportional to the concentration of hydrogen peroxide, whereas the ability of the bubbles to escort the gaseous hydrogen fluoride is linked to the S/L ratio, which is clearly shown in Figure 7.

$$2H_2O_{2(aq)} \rightarrow 2H_2O_{(l)} + O_{2(g)} \quad (18)$$

We can easily explain the beneficial effect of hydrogen peroxide on the Al leaching degree using the corrosion approach (Figure 7). As a result of its high oxidation potential in an acidic environment, hydrogen peroxide is an effective etching agent for both aluminum (19) and the protective film (20). This is supported by the long standing practice of etching the alumina surface with hydrogen peroxide to prepare catalyst carriers [50].

$$2Al_{(s)} + 3H_2SO_{4(aq)} + 3H_2O_{2(aq)} \rightarrow Al_2(SO_4)_{3(aq)} + 6H_2O_{(l)} \quad (19)$$

$$Al_2O_{3(s)} + 3H_2SO_{4(aq)} + 2H_2O_{2(aq)} \rightarrow Al_2(SO_4)_{3(aq)} + 5H_2O_{(l)} + O_{2(g)} \quad (20)$$

As we saw in previous experiments, the leaching degree of Zn reaches a plateau of 40% due to the exhaustion of available zinc oxide (14), whereas zinc sulfide passivates (15). However, the presence of hydrogen peroxide disrupts zinc sulfide passivation by causing it to produce zinc sulfates without the formation of a sulfur layer (21). Instead, hydrogen peroxide produces hydrogen sulfide gas [51], which leaves the leaching solution and, as seen in Figure 7, clearly promotes an increase in the Zn leaching degree.

$$2ZnS_{(s)} + 4H_2O_{2(aq)} + H_2SO_{4(aq)} \rightarrow 2ZnSO_{4(aq)} + H_2S_{(g)} + 4H_2O_{(l)} \quad (21)$$

Copper leaching is also assisted by the presence of strong oxidizers, so the leaching degree in the presence of hydrogen peroxide is always higher than in its absence (Figure 7). The reaction of copper leaching with sulfuric acid in the presence of hydrogen peroxide is given by Equation (22).

$$Cu_{(s)} + H_2SO_{4(aq)} + H_2O_{2(aq)} \rightarrow CuSO_{4(aq)} + 2H_2O_{(l)} \quad (22)$$

As previously stated, cobalt leaching is strongly related to the presence of oxidative media, and as it shown in Figure 7, when the S/L ratio is low (which means cobalt leaching is affirmative), the presence of another oxidation agent negates the cobalt leaching. When the S/L ratio is increased and the overall leaching rate is a diffusion limited process, the presence of hydrogen peroxide is positive, because this oxidation agent is also involved in oxidation of the polymer matrices. This improves both the transfer of cobalt to the leachate bulk and reduces oxidation agents. Cobalt is only in a +2 oxidation state after leaching

because it is transformed from the unsTable 3+ ion with high oxidation potential. Reactions of cobalt and lithium leaching with sulfuric acid is shown in Equation (3).

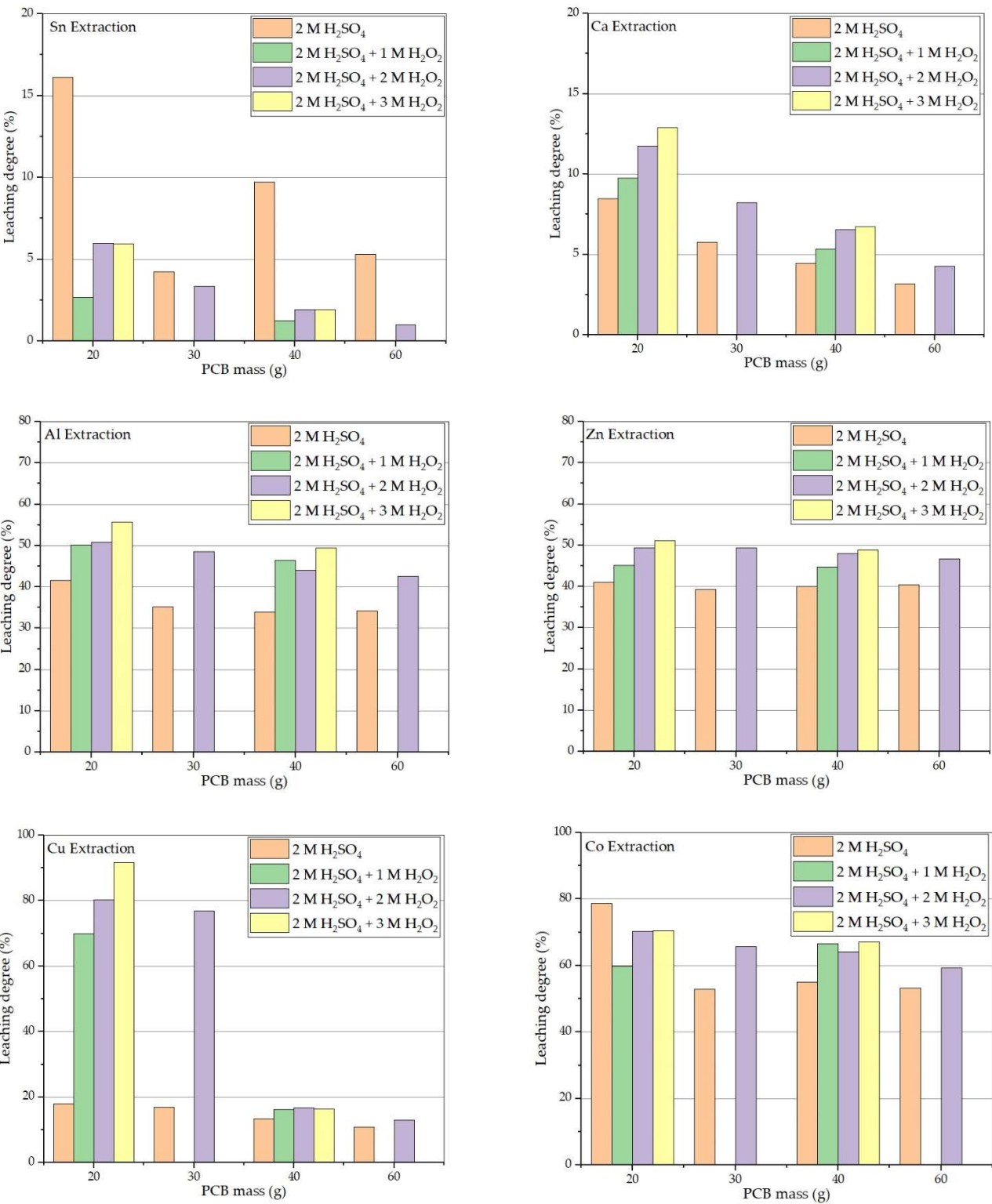

**Figure 7.** Element extraction after 4 h leaching in 200 mL 2 M sulfuric acid with additions of different concentrations of hydrogen peroxide (1 M 23.46 mL), (2 M 46.92 mL), and (3 M 70.38 mL) for four starting amounts of fine PPCB powders.

**Table 3.** Leaching degree for NADES leaching without and with hydrogen peroxide, respective ORP of leaching solution included.

|  | ORP (mV) | Time (h) | Leaching Degree (%) | | | | | | | | | | |
|---|---|---|---|---|---|---|---|---|---|---|---|---|---|
|  |  |  | Al | Cu | Zn | Sn | Nd | Pd | Ca | Pb | Ba | Co | Pt |
| without H$_2$O$_2$ | −388.5 | 2 | 16.9 | 1.6 | 25.6 | 3.6 | n.d. | n.d. | 0.3 | 26.2 | 0.1 | n.d. | n.d. |
|  | −375.6 | 4 | 15.9 | 2.5 | 28.9 | 1.5 | n.d. | n.d. | 0.3 | 27.1 | 0.5 | 27.6 | 22.7 |
|  | −389.3 | 6 | 14.8 | 5.1 | 28.9 | 1.1 | n.d. | n.d. | 0.2 | 21.0 | 1.0 | 29.0 | n.d. |
|  | −418.7 | 8 | 12.5 | 4.5 | 24.3 | 0.8 | n.d. | n.d. | 0.1 | 14.4 | 1.0 | 25.7 | n.d. |
|  | ORP | Time (h) | Leaching Degree (%) | | | | | | | | | | |
|  |  |  | Al | Cu | Zn | Sn | Nd | Pd | Ca | Pb | Ba | Co | Pt |
| with H$_2$O$_2$ | −317.1 | 2 | 23.5 | 21.9 | 39.3 | 4.3 | 19.9 | n.d. | 2.0 | 28.0 | 1.8 | 48.3 | 98.2 |
|  | −277.3 | 4 | 6.8 | 17.2 | 29.2 | 3.2 | 11.9 | 5.6 | 0.5 | 17.1 | 0.8 | 44.7 | 95.3 |
|  | −230.4 | 6 | 2.2 | 33.7 | 25.8 | 3.3 | 12.6 | 5.8 | 0.4 | 12.5 | 1.0 | 40.9 | 86.3 |
|  | −83.5 | 8 | 0.6 | 12.9 | 16.1 | 1.8 | / | n.d. | 0.2 | 7.3 | 0.4 | 25.7 | 27.5 |

### 3.5. Acid Mine Drainage and Distilled Water Leaching with the Aid of Hydrogen Peroxide

The pH of AMD (pH = 2.01) indicates that the acid concentration, mostly sulfuric [52], is two orders of magnitude lower than in leaching solutions that used commercial sulfuric acid. Thus, AMD cannot leach as well as commercial sulfuric acid. However, different hydrogen peroxide concentrations were added with and without concentrated sulfuric acid to test AMD as a leaching medium. The lowest and highest hydrogen peroxide concentrations were added to distilled water to study AMD's baseline effect, as seen in Figure 8. In distilled water, the influence of hydrogen peroxide is most evident in Co leaching, since Co$^{3+}$ ions are used up by catalytically assisting the decomposition of hydrogen peroxide (18) in electron transfer [53]. Hydrogen peroxide oxidizes zinc sulfide, slightly increasing Zn leaching [43]. Calcium leaching rises 2% regardless of hydrogen peroxide concentration. Despite low Co concentration in the PPCB (Table 1), and the fact that AMD is acidic enough to allow Co$^{3+}$ ions to participate in redox reactions, Co leaching with AMD is always high (above 50%). The effects of AMD and hydrogen peroxide leaching on zinc are similar. Finally, using the corrosion approach, we can explain the observed differences in aluminum and copper based on their starting concentration in AMD (Table 2) and its pH of 2.01. In conclusion, AMD is suitable for selective leaching with limited applications.

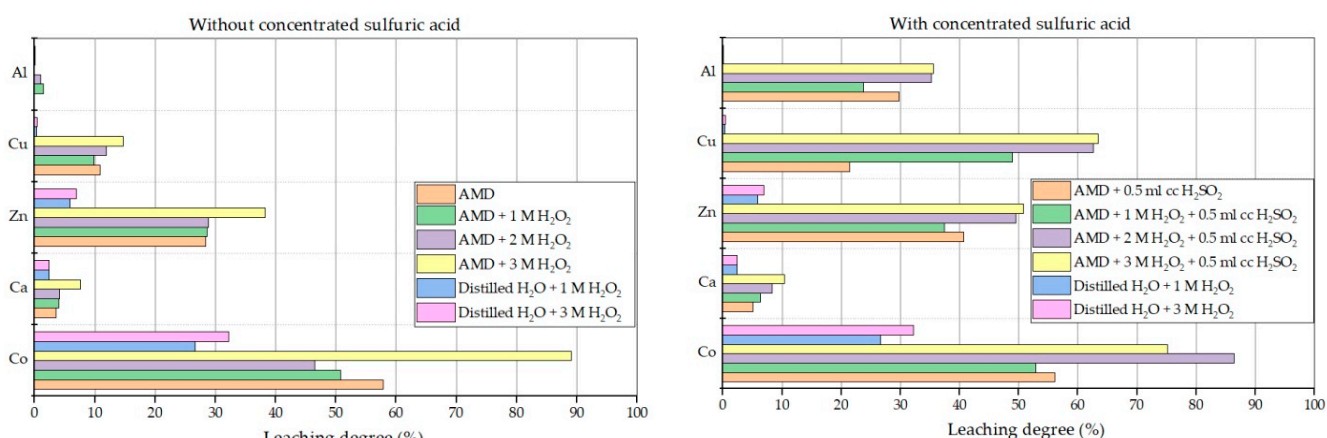

**Figure 8.** Influence of hydrogen peroxide on AMD leaching without and with concentrated sulfuric acid with reference to hydrogen peroxide distilled water leaching of fine PPCB powder.

### 3.6. Leaching with Glycine Based Lixiviant (NADES)

Sodium glycinate, a natural deep eutectic solvent (NADES), is used to extract valuable metals like Cu, Au, Pd, Pt, and Ag from many raw materials, including PPCB. In gold ore leaching, glycinate (Gly) anions form more stable complexes with noble metals compared

to Fe, Mn, Cr, Mg, Al, and Si [54]. Gly-based leaching agents are environmentally friendly and renewable. Our experiment adjusted the glycinate solution's pH to above 12 with sodium hydroxide. Table 3 shows that the absence of hydrogen peroxide prevented sodium glycinate (NaGly) from effectively leaching copper and palladium. Sodium hydroxide alone can leach amphoteric metals like Al, Zn, Pb, and Sn and Cu as hydroxo complexes. Equations (23)–(26) for Zn, Pb, Sn, and Cu show sodium hydroxide leaching reactions, while Al was discussed earlier in Equation (10).

$$ZnO_{(s)} + 2NaOH_{(aq)} + H_2O_{(l)} \rightarrow Na_2[Zn(OH)_4]_{(aq)} \tag{23}$$

$$2Pb_{(s)} + O_{2(g)} + 4NaOH_{(aq)} + 2H_2O_{(l)} \rightarrow 2Na_2[Pb(OH)_4]_{(aq)} \tag{24}$$

$$Sn_{(s)} + 2NaOH_{(aq)} + O_{2(g)} + 2H_2O_{(l)} \rightarrow Na_2[Sn(OH)_6]_{(aq)} \tag{25}$$

$$2Cu_{(s)} + O_{2(g)} + 4NaOH_{(aq)} + 2H_2O_{(l)} \rightarrow 2Na_2[Cu(OH)_4]_{(aq)} \tag{26}$$

Leaching with NaGly without hydrogen peroxide steadily decreases the oxidation reduction potential (ORP), while adding hydrogen peroxide hourly increases the ORP. Table 3 shows that hydrogen peroxide increases leaching for neodymium, as well as amphoteric and noble metals. Equations (27)–(31) show how hydrogen peroxide improves NaGly leaching, and numerous studies have been published with similar results [33,35,55].

$$2Cu_{(s)} + 4NH_2CH_2COONa + 2H_2O_{2(aq)} \rightarrow 2Cu(NH_2CH_2COO)_{2(aq)} + 4NaOH_{(aq)} \tag{27}$$

$$2Au_{(s)} + 4NH_2CH_2COONa + H_2O_{2(aq)} \rightarrow 2Na\left[Au(NH_2CH_2COO)_{2(aq)}\right]_{(aq)} + 2NaOH_{(aq)} \tag{28}$$

$$Nd_2Fe_{14}B_{(s)} + 6NH_2CH_2COONa + 3H_2O_{2(aq)} \rightarrow 2Nd(NH_2CH_2COO)_{3(aq)} + 6NaOH_{(aq)} + B_{(s)} + 14Fe_{(s)} \tag{29}$$

$$Pt_{(s)} + 2NH_2CH_2COONa + H_2O_{2(aq)} \rightarrow Pt(NH_2CH_2COO)_{2(aq)} + 2NaOH_{(aq)} \tag{30}$$

$$Pd_{(s)} + 2NH_2CH_2COONa + H_2O_{2(aq)} \rightarrow Pd(NH_2CH_2COO)_{2(aq)} + 2NaOH_{(aq)} \tag{31}$$

We can assume that the presence of pyrolyzed polymer residue acts as redox agent source for already oxidized or complex-bound metals. The pyrolyzed polymers residue is high in carbon, and carbon is known for its reducing properties [54]. As a result, as shown in Table 3, the leaching rate decreases as contact time increases. It can be concluded that leaching with NaGly proceeds as follows:

1. Oxidation of metals present in PPCB with readily available hydroxyl radical (when $H_2O_2$ is present in solution) or superoxide and other reactive oxygen species (in the absence of $H_2O_2$ but in presence of dissolved oxygen in leaching media);
2. Bonding the oxidized metal in a complex with a glycinate anion (formation of metal–glycinate complexes);
3. Metal cementation from metal–glycinate complexes in the presence of easily accessible pyrolyzed polymer residue (that are carbon rich materials).

### 3.7. Two-Pronged Leaching without Pretreatment

The two-pronged leaching without pretreatment was conducted in sulfuric and nitric acids for four hours each. The extraction mechanisms of most metals (Cu, Al, Co, Ca, Zn, and Sn) and the S/L ratio of 0.1 were previously discussed. This experiment differs in two ways. First, the system was open, with circulating air that requires water to be added to maintain leaching solution volume. Second, the leaching solution was heated in a water bath under ultrasound sonification. We can see that the ORP increases in both leaching cases using sulfuric or nitric acid (Table 4), although the increase is more noticeable for sulfuric acid because the leaching solution contains a higher concentration of oxidized metals. The two-pronged leaching approach is unique in that precipitates formed in the first leaching medium can be dissolved in the second.

**Table 4.** Two-pronged leaching parameters and element leaching degree without pretreatment.

| Leaching Agent | ORP (mV) | Time (h) | Leaching Degree (%) | | | | | | | | | | |
|---|---|---|---|---|---|---|---|---|---|---|---|---|---|
| | | | Al | Cu | Zn | Sn | Nd | Pd | Ca | Pb | Ba | Co | Pt |
| Sulfuric Acid | 336.2 | 2 | 30.3 | 53.6 | 53.1 | 1.5 | 73.8 | n.d. | 12.9 | 0.5 | n.d. | 94.4 | n.d. |
| | 416 | 4 | 38.5 | 77.6 | 54.2 | 1.7 | 61.5 | 4.9 | 10.9 | 0.6 | 0.1 | 93.8 | 22.7 |
| Nitric acid | 874.2 | 2 | 9 | 17.5 | 6.1 | 2 | 17.6 | 23.7 | 12.2 | 23.4 | n.d. | n.d. | n.d. |
| | 891.4 | 4 | 13.4 | 14.3 | 4.1 | 1.3 | 15.8 | 33 | 19.6 | 49.5 | 5.4 | n.d. | n.d. |
| Summed LD * (4 h + 4 h) | | | 51.9 | 91.9 | 58.3 | 3.0 | 77.3 | 37.9 | 30.5 | 50.1 | 5.5 | 93.8 | 22.7 |

* LD = Leaching degree.

One of the metals that was not previously discussed is neodymium (Nd), whose presence in PCBs is contributed to permanent "Neo magnets", a ferrous alloy $Nd_2Fe_{14}B$. Table S1 in the Supplementary Material shows the degree of Nd leaching at low S/L ratios. Regardless of whether the leaching solution is sulfuric or nitric acid, Nd leaching forms a passive protective layer of B, as shown by Equations (32) for sulfuric and (33) for nitric acid. This is why in Table 4 we see a decrease in leaching degree due to passivation over time. Similar to cobalt leaching in sulfuric acid from lithium cobalt oxide ($LiCoO_2$) presented in Equation (3), the leaching of cobalt and lithium in nitric acid is presented in Equation (34).

$$Nd_2Fe_{14}B_{(s)} + 17H_2SO_{4(aq)} \longrightarrow Nd_2(SO_4)_{3(aq)} + 14FeSO_{4(aq)} + 17H_{2(g)} + B_{(s)} \quad (32)$$

$$Nd_2Fe_{14}B_{(s)} + 48HNO_{3(aq)} \longrightarrow 2Nd(NO_3)_{3(aq)} + 14Fe(NO_3)_{3(aq)} + 24H_{2(g)} + B_{(s)} \quad (33)$$

$$LiCoO_{2(s)} + (2+y)HNO_{3(aq)} + M \rightarrow LiNO_{3(aq)} + Co(NO_3)_2 + M(NO_3)_y + (1+0.5y)H_2O_{(l)} \quad (34)$$

Almost all of the cobalt was leached in sulfuric acid; therefore, this reaction did not occur. However, for calcium this is not the case because the expenditure of hydrofluoric acid is the limiting factor for Ca leaching (Equation (5)). There are no discernible differences in calcium fluoride leaching in sulfuric acid (Equation (5)) and nitric acid (Equation (35)), but there is a difference in the Ca leaching degree in Table 4. This is because nitric acid leaching generates gasses like NO and $NO_2$ in reactions with other elements, and these gases function similarly to gas bubbles. We discussed how hydrogen peroxide decomposes to produce oxygen (Equation (18)), escorting gaseous hydrofluoric acid out of the solution, earlier in Section 3.4, and this is even more pronounced with nitrous gasses due to the 2.5 L per minute injection of air.

$$CaF_{2(s)} + 2HNO_{3(aq)} \leftrightarrows Ca(NO_3)_{2(aq)} + 2HF_{(aq)} \quad (35)$$

Unlike in sulfuric acid, where Sn can be extracted from fine PPCB powder in the form of soluble product $SnSO_4$ (Equation (6)) and insoluble product $SnO_2 \cdot xH_2O$ (Equation ((7)), the only product that can be extracted in nitric acid is an insoluble one (Equations (36) and (37)). As discussed in Section 3.2, since the sulfuric acid concentration was 1 M and the S/L ratio was 0.1, the leaching degree of Sn was very low.

$$6Sn_{(s)} + 4HNO_{3(aq)} + 5O_{2(g)} + (6x-2)H_2O_{(l)} \longrightarrow 6SnO_2 \cdot xH_2O_{(s)} + 4NO_{2(g)} \quad (36)$$

$$3Sn_{(s)} + 4HNO_{3(aq)} + 4H_2O_{2(aq)} + (3x-6)H_2O_{(l)} \longrightarrow 3SnO_2 \cdot xH_2O_{(s)} + 4NO_{2(g)} \quad (37)$$

In Section 3.1, the corrosion approach is used to explain aluminum leaching, and the influence of nitric acid is shown in reaction (9). The degree of Al leaching increases steadily in both sulfuric and nitric acid mediums (Table 4). It is more noticeable in sulfuric acid because it was used first and had more Al readily available, but the total leaching degree is greater than 50%. The main difference between sulfuric and nitric acid in copper leaching is in the leaching product. In both leaching solutions, copper can be leached with the

expenditure of dissolved oxygen (sulfuric acid–reaction 13; nitric acid–reaction 39) and without it (sulfuric acid–reaction 12; nitric acid–reaction 38).

However, the product of Cu leaching in nitric acid, copper (II) nitrate, can hydrolyze according to reaction (40). This explains the drop in the Cu leaching degree seen in Table 4, present for nitric acid leaching but absent for sulfuric acid.

$$3Cu_{(s)} + 8HNO_{3(aq)} \rightarrow 3Cu(NO_3)_{2(aq)} + 4H_2O_{(l)} + 2NO_{(g)} \tag{38}$$

$$2Cu_{(s)} + 4HNO_{3(aq)} + O_{2(g)} \rightarrow 2Cu(NO_3)_{2(aq)} + 2H_2O_{(l)} \tag{39}$$

$$8Cu(NO_3)_{2(aq)} + 6H_2O_{(l)} \longrightarrow 4Cu_2(NO_3)(OH)_{3(s)} + 12NO_{2(g)} + 3O_{2(g)} \tag{40}$$

As we discussed in Section 3.1, PPCB contains zinc in two forms: zinc oxide and zinc sulfide. The leaching of zinc oxide and sulfide in sulfuric acid are presented in reactions (14 and 15), while the leaching of zinc oxide and sulfide in nitric acid are presented in reactions (41 and 42). Despite the fact that the Zn reactions in sulfuric and nitric acids are analogous, the main difference is in the nature of leaching product. Another noticeable difference from Section 3.1 is that due to the effect of ultrasound sonification, the zinc leaching plateau in sulfuric acid is raised to around 54%.

$$ZnO_{(s)} + 2HNO_{3(aq)} \rightarrow Zn(NO_3)_{2(aq)} + H_2O_{(l)} \tag{41}$$

$$3ZnS_{(s)} + 8HNO_{3(aq)} \rightarrow 3Zn(NO_3)_{2(aq)} + 2NO_{(g)} + 3S_{(s)} + 4H_2O_{(l)} \tag{42}$$

The leaching degree of Zn decreased in nitric acid over time (Tables 4 and 5), due to the precipitation of zinc (II) nitrate with iron in reaction (43) producing a spinel oxide that is inert toward acidic media. However, unlike for copper, nitric acid has a beneficial effect on the leaching of zinc. This is due to the ability of nitric acid to dissolve the passive layer of sulfur that forms over zinc sulfide by reaction (42). The reaction of dissolving sulfur with nitric acid consequently produces sulfuric acid and is illustrated by Equation (44).

$$Zn(NO_3)_{2(aq)} + 2Fe(OH)_{3(s)} \rightarrow ZnFe_2O_{4(s)} + 2HNO_{3(aq)} + 2H_2O_{(l)} \tag{43}$$

$$S_{(s)} + 2HNO_{3(aq)} \rightarrow H_2SO_{4(aq)} + 2NO_{(g)} \tag{44}$$

### 3.8. Leaching with Two Strong Oxidizing Agents (Nitric Acid and Hydrogen Peroxide)

The novel idea was to investigate the synergetic effect of two strong oxidizing agents on fine PPCB powder leaching for various metals. Nitrates are one of the most numerous soluble salts known in inorganic chemistry, and for that reason, nitric acid could efficiently leach several metals that other mineral acids could not (such as lead, palladium, platinum, seen in Table 5).

**Table 5.** Influence of hydrogen peroxide on nitric acidic leaching parameters.

| ORP (mV) | Time (h) | Leaching Degree (%) | | | | | | | | | | |
|---|---|---|---|---|---|---|---|---|---|---|---|---|
| | | Al | Cu | Zn | Sn | Nd | Pd | Ca | Pb | Ba | Co | Pt |
| 779.5 | 2 | 46.8 | 92.6 | 76.3 | 1.2 | 94.3 | 26.5 | 30.1 | 65.5 | 20.9 | 99.2 | 87.3 |
| 770.7 | 4 | 41.9 | 94.7 | 74.5 | 0.8 | 93.7 | 64.3 | 37.1 | 78.8 | 22.9 | 95.1 | 93.4 |
| 798.2 | 6 | 48.7 | 98.9 | 70.1 | 0.8 | 86.4 | 87.9 | 29.8 | 81.9 | 26.1 | 92.7 | 98.1 |
| 859.4 | 8 | 36.7 | 71.9 | 54.3 | 0.4 | 75.0 | 97.7 | 29.8 | 94.9 | 19.7 | 80.7 | 99.6 |

Table 4 shows nitric acid leaching as a second leaching agent after most metals were extracted, and Table 6 shows it as a first leaching agent after some or none of the metals were extracted. Table 5 shows the full effect of nitric acid leaching with hydrogen peroxide. Section 3.7 compared sulfuric acid and nitric acid leaching of Al, Cu, Zn, Sn, Co, Ca, and Nd. Hydrogen peroxide barely affects Al, Co, Sn, and Cu leaching, and their reaction

equations are in the Supplementary Materials (Table S2.). Oxygen bubbles do not help remove hydrofluoric gas produced by Ca leaching due to the emission of nitrous gasses.

Reaction (45) affects boron oxidation only in hot concentrated nitric acid, but reaction (46) dominates in the presence of hydrogen peroxide. Both reactions decompose the passive protective layer of B over Neo magnets in nitric acid, reaction (33). This effect is most noticeable in the first four hours of leaching when the Nd leaching degree is above 90%, but due to the expenditure of nitric acid on other metals and mass transfer limitation, passivation prevails over time.

$$B_{(s)} + HNO_{3(aq)} + H_2O_{(l)} \rightarrow H_3BO_{3(aq)} + NO_{(g)} \tag{45}$$

$$5B_{(s)} + 3HNO_{3(aq)} + 3H_2O_{2(aq)} + 3H_2O_{(l)} \rightarrow 5H_3BO_{3(aq)} + 3NO_{(g)} \tag{46}$$

When the temperature of the nitric acid leaching solution is raised to temperatures above 40 °C, the lead present in the fine PPCB powder is easily oxidized to lead (II) nitrate by reaction (47). The addition of hydrogen peroxide to the leaching solution alters the Pb oxidation reaction (48). The one difference in the effect of hydrogen peroxide is that, even though the overall leaching degrees were above 80%, the leaching degree steadily drops in Table 6, whereas it rises in Table 5 with hydrogen peroxide.

$$3Pb_{(s)} + 8HNO_{3(aq)} \rightarrow 3Pb(NO_3)_{2(aq)} + 2NO_{(g)} + 4H_2O_{(l)} \tag{47}$$

$$5Pb_{(s)} + 12HNO_{3(aq)} + 2H_2O_{2(aq)} \rightarrow 5Pb(NO_3)_{2(aq)} + 2NO_{(g)} + 8H_2O_{(l)} \tag{48}$$

Barium presence in the PPCB can be contributed to its two-fold applications. Barium sulfate is present as a fire retardant while barium strontium titanate, as a cathode ray tube material [56]. It is widely known that barium sulfate is a barium salt with very low solubility (Ksp = $1.1 \times 10^{-10}$ [47]); therefore, barium leaching from this salt is very limited. On the other hand, barium ceramics can be leached with nitric acid according to reaction (49), since barium nitrate is a water-soluble salt and, as it can be seen in Tables 5 and 6, barium leaching rates achieved are about 30%.

$$Ba_xSr_{1-x}TiO_{3(s)} + 2HNO_{3(aq)} \rightarrow xBa(NO_3)_{2(aq)} + (1-x)Sr(NO_3)_{2(aq)} + TiO_{2(s)} + H_2O_{(l)} \tag{49}$$

In Section 3.6, we discussed how pyrolyzed polymer residue cancels out any noble metal leaching conducted by Gly media, by reducing the metals in complexes and precipitating them in elemental form. Experiments conducted with two strong oxidizing agents have shown that such unfavorable actions can be avoided, and platinum group metals can be leached successfully to a degree of almost 100%. Nitrate salts formed in reactions (50, 51) are stable, and due to the presence of nitric acid, Pd (II) is not able to be reduced to its elemental form again. Similar behavior can be seen in the case of platinum (Table 5).

$$3Pd_{(s)} + 8HNO_{3(aq)} \longrightarrow 3Pd(NO_3)_{2(aq)} + 2NO_{(g)} + 4H_2O_{(l)} \tag{50}$$

$$5Pd_{(s)} + 12HNO_{3(aq)} + 2H_2O_{2(aq)} \longrightarrow 5Pd(NO_3)_{2(aq)} + 2NO_{(g)} + 8H_2O_{(l)} \tag{51}$$

*3.9. Two-Pronged Leaching with Two Pretreatment Approaches*

Two pretreatments were investigated and compared in this work in order to discover their impact on PPCB leaching. One is "chemical pretreatment" closed system, with eight-hour NaOH leaching at 70 °C under ultrasound sonification, while the other is "physical pretreatment", with trichloroethylene swelling of pyrolyzed polymer residue. The primary goal of the sodium hydroxide treatment is to etch (reaction (52)) and leach fiberglass needles present in the fine PPCB powder in order to liberate more metals to be extracted. A secondary benefit of this pretreatment is the insight into the sodium hydroxide leachability separate from glycinate media. On the other hand, trichloroethylene does not extract any

metals since it only alters the pyrolyzed polymer residue and is evaporated rapidly in the process [57].

$$ySiO_{2(s)} + 2xNaOH_{(aq)} \longrightarrow (Na_2O)_x(SiO_2)_{y(gel)} + xH_2O_{(l)} \tag{52}$$

Furthermore, after both pretreatments, two-pronged leaching was carried out, but this time the first leaching agent used was nitric acid while the second was NaGly with the addition of hydrogen peroxide (Table 6). This is important since here we can see how nitric acid behaves as a leaching agent without the influence of hydrogen peroxide.

**Table 6.** Influence of pretreatments on two-pronged (Nitric–NaGly) leaching.

| Element | Sodium Hydroxide Treatment | NaOH Pretreatment | | | | Swelling Pretreatment | | | |
|---|---|---|---|---|---|---|---|---|---|
| | | Nitric Acid Leaching | | NaGly Leaching | | Nitric Acid Leaching | | NaGly Leaching | |
| | | ORP (mV) | | | | ORP (mV) | | | |
| | | 865.7 | 851.7 | −335 | −306.3 | 866 | 854.4 | −243 | −291 |
| | | Time (h) | | | | Time (h) | | | |
| | | 2 | 4 | 2 | 4 | 2 | 4 | 2 | 4 |
| | | Leaching Degree (%) | | | | | | | |
| Al | 23.1 | 0.031 | 0.030 | n.d. | n.d. | 39.8 | 43.9 | n.d. | n.d. |
| Cu | 6.0 | 86.6 | 77.8 | 0.1 | 0.5 | 93.4 | 95.8 | 1.1 | 0.3 |
| Zn | 33.4 | 36.4 | 29.8 | 0.3 | 0.3 | 68.3 | 65.3 | 1.9 | 0.3 |
| Sn | 11.3 | 7.6 | 6.7 | 0.1 | n.d. | 1.4 | 2.4 | 0.1 | 0.1 |
| Nd | n.d. | 92.2 | 78.4 | n.d. | n.d. | 95.3 | 93.8 | n.d. | n.d. |
| Pd | n.d. | 6.9 | 5.7 | 22.5 | 25.3 | 27.3 | 27.5 | 45.9 | 31.1 |
| Ca | 0.03 | 55.0 | 53.1 | 0.2 | 0.5 | 31.8 | 24.5 | 0.1 | 0.2 |
| Pb | 59.3 | 41.5 | 29.4 | n.d. | n.d. | 97.1 | 86.9 | 0.4 | n.d. |
| Ba | n.d. | 24.2 | 34.3 | n.d. | n.d. | 23.3 | 21.6 | 0.7 | n.d. |
| Co | n.d. | 97.7 | 93.8 | n.d. | n.d. | 99.5 | 99.1 | n.d. | n.d. |
| Pt | n.d. | 88.3 | 94.2 | n.d. | n.d. | 89.1 | 93.2 | n.d. | n.d. |

Sodium hydroxide reacts with amphoteric metals Al, Zn, Pb, and Sn, which were extracted during pretreatment (Equations (10), (23)–(25)). Since tin (IV) oxide is acidic and cannot be prevented from forming, tin extraction is the lowest (reaction 25). Oxygen diffusion is needed for copper extraction, and sodium hydroxide alone cannot oxidize copper. Reaction (26) is diffusion-controlled because oxygen is required on the copper surface. Sodium hydroxide increases Ca leaching. As sodium hydroxide etches $SiO_2$, hydrofluoric acid is consumed faster, moving the equilibrium in reaction (4) forward. As mentioned, barium strontium titanate reacts with nitric acid to form soluble nitrate salts (49). Table 6 shows that pretreatments do not significantly affect the degree of Ba leaching.

Water rinsing of the PPCB after pretreatment and before leaching in nitric acid is linked to the difference in the Al leaching degree in two-pronged leaching with NaOH pretreatment. The sodium aluminate formed by reaction (10) interacts with the water used to rinse sodium hydroxide. The sodium aluminate forms a passive layer of aluminum hydroxide according to reaction (53). This additional passive layer inhibits Al, preventing nitric acid from extracting it, as shown in Table 6.

$$NaAlO_{2(aq)} + 2H_2O_{(l)} \rightarrow Al(OH)_{3(s)} + NaOH_{(aq)} \tag{53}$$

For the swelling pretreatment in nitric acid leaching, a positive influence on the leaching degree can be seen for metals such as Cu, Co, and Nd (Table 6). This can be contributed to the swelling of pyrolyzed polymer residue that enables easier and more effective penetration of leaching agents into the PPCB media.

However, two-pronged leaching (Sulfuric–Nitric) without pretreatment is the same as two-pronged leaching (Nitric–NaGly) with pretreatment. Tables 4 and 6 show that the second leaching agent had poor metal leachability since most metals were extracted during

the first leaching procedure. Two-pronged leaching did not selectively leach metal species in fine PPCB powders regardless of the pretreatment.

## 4. Conclusions

In our overall multifocal approach to element extraction from fine PPCB powder we found that:

- The aluminum leaching degree is limited by the increase in the S/L ratio since it is diffusion controlled. The increase of sulfuric acid concentration lowers oxygen solubility, and this promotes Al leaching. As expected, with the increase in temperature, the Al leaching degree increases, as per the corrosion approach. The addition of hydrogen peroxide in an acidic environment acts a potent etching agent for both aluminum and aluminum oxide, but in the alkali environment it hydrolyses sodium aluminate forming aluminum hydroxide.
- The copper leaching degree is obstructed with an increase in S/L ratio, similar to Al, since it is also diffusion controlled. However, an increase in sulfuric acid concentration that lowers oxygen solubility is detrimental to the Cu leaching degree, while the increase in leaching temperature is very beneficial. Additionally, the presence of an oxidation agent, be it air or hydrogen peroxide, significantly promotes Cu leaching, and this is even more pronounced under ultrasound sonification. For leaching in nitric acid this is not needed, but there is a drawback as copper (II) nitrate hydrolyses overtime. Without the presence of hydrogen peroxide, Cu leaching in an alkali environment is insignificant.
- The cobalt leaching degree is proportional to the reduction of $Co^{3+}$; therefore, it is strongly related to mass transfer and the materials present in the PPCB that can be oxidized. This is why the presence of hydrogen peroxide lowers the Co leaching degree. However, it is highly dependent on the penetration of the leaching agent into the PPCB matrix, which can be promoted with ultrasound and swelling. Both sulfuric and nitric acid are very efficient in leaching Co, but the leaching in NADES is limited to the formation of cobalt glycinate, while in pure alkali media there is no leaching at all.
- The zinc leaching degree is limited by the expenditure of zinc sulfide that passivizes easily in sulfuric acid, and is elevated by a temperature increase or ultrasound sonification. Nitric acid dissolves the passive layer of sulfur, while the addition of hydrogen peroxide to sulfuric acid produced hydrogen sulfide gas instead of sulfur. The overall leaching degree of Zn is attributed to the leaching of Zn sulfide and Zn oxide, both of which are present in PPCB.
- The tin leaching degree is limited by two concurrent reactions: the production of soluble Sn (II) salts and the production of insoluble Sn (IV) compounds. At low S/L ratios, there is a critical sulfuric acid concentration between 1 M and 2 M, below which Sn (IV) oxidation is promoted. However, in nitric acid there are no concurrent reactions and only insoluble Sn (IV) compounds are produced.
- The calcium leaching degree is closely linked to the way that produced hydrofluoric acid leaves the leaching solution. Hydrofluoric acid can be either spent on fiberglass dissolution or escorted in gaseous form from the leaching solution. Fiberglass dissolution is promoted by sodium hydroxide while gas migration is promoted either by adding hydrogen peroxide or by using nitric acid.
- The neodymium leaching degree is favorable in acidic media while unfavorable in alkali media without the presence of hydrogen peroxide. The palladium leaching degree is promoted in a strong oxidation environment. This goes for platinum leaching degree as well; however, it can also be achieved in a NADES leaching solution. The lead leaching degree is the highest in nitric acid but also promoted in an alkali environment. Finally, the barium leaching degree was only seen to be promoted in nitric acid.

For NADES leaching, all the extracted metal gradually cemented over time in the solution, due to their reduction with the pyrolyzed polymer residue present in the fine PPCB powder.

Regardless of the pretreatment or the sequence of leaching media applied, consecutive two-pronged leaching cannot be used for selective metal extraction.

Acid mine drainage was found to be suitable for selective leaching with very limited application.

**Supplementary Materials:** The following supporting information can be downloaded at: https://www.mdpi.com/article/10.3390/met12122021/s1, Element origin list; Table S1: Leeching degree after 4 h; Table S2: Leaching with two strong oxidizing agents.

**Author Contributions:** Conceptualization, G.J. and M.B.; funding acquisition, M.S. and B.F.; formal analysis, B.M. and S.S.; investigation, N.P., G.J. and M.B.; methodology, N.P. and S.R.S.; supervision, M.S. and B.M.; writing—original draft, N.P., M.B., G.J., B.M. and S.R.S.; writing—review and editing, M.S. and S.S. All authors have read and agreed to the published version of the manuscript.

**Funding:** This work was supported by the Ministry of Education, Science and Technological Development of the Republic of Serbia (Grant No. 451-03-68/2022-14/200023). The authors would like to thank the Ministry of Education, Science and Technological Development of the Republic of Serbia and DAAD, Germany, for funding of the Project No.: 57513134.

**Institutional Review Board Statement:** Not applicable.

**Informed Consent Statement:** Not applicable.

**Data Availability Statement:** Not applicable.

**Conflicts of Interest:** The authors declare no conflict of interest.

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
