# Peer review of "A Multifocal Study Investigation of Pyrolyzed Printed Circuit Board Leaching"

_metals, doi:10.3390/met12122021_

Round 1
Reviewer 1 Report
The paper describes a recycling process in which valuable metals are recovered from end-of-life printed circuit board (PCB) by leaching them with various reagents. However, the authors have added too many reagents into the system which is already complex. The paper lacks coherency in its objectives and scientific merits in delineating the chemistry of the system. One fatal failure of the paper is its poor characterization of chemistry of the system. For example, the chemical state of metal elements such as metal, alloy, oxide, etc. is not clearly defined. Reactions used throughout the text are based mostly on pure metal elements which may not be relevant and representing the system. The most of chemical reactions given in the paper are speculative and taken from the literature without justification to the specifics of the system at hand.
The novelties of the paper are using the pyrolyzed grinding and using the acid mine drainage (AMD) as a leaching reagent but failed to present some of their inherent drawbacks. For example, during pyrolysis, some of the plastics might have been burnt to produce carbon. If this happens, it may have a deleterious effect such as adsorption of metal ions from the leach liquor. Adding AMD means adding dirty solution into the leach liquor; therefore, it may add a difficult problem in recovering individual metals from the leach liquor. These and other problems associated with such processes should have been discussed thoroughly.
Only a portion of ground PCB under controlled atmosphere at a high temperature was used and therefore, the overall recovery of metal values cannot be evaluated. The chemistry of the ground materials has been speculated but no experimental evidence is given. It is unclear what percentage of cobalt is associated with Li-Co batteries. For example, cobalt is used in the circuit board as part of wire, co-precipitant with gold, and as part of Li-battery. Most of leaching was carried out at 60°C without any reason for it. Is it because of the tolerable temperature range restriction by DES? Please explain. The chemistry of the leaching reaction of Ca, Ba, Al that are of no economic value as a recycling product but the relevance to the chemistry of valued metals such as Co, Li, precious metals is poorly examined.
About the results given in Fig 2, there are so many variables involved in this study and this figure represents one of hundreds possible scenarios. The leaching efficiency given in terms of the S/L ratio as a variable is vague and difficult to understand.
The English grammar of the text should be checked carefully.
Lines 71-74: confusing and check the grammar
Lines 76-78: confusing
Line 89-91: Unqualified and irresponsible statement. No lixiviant is qualified to be the best reagent since the leaching rate is case specific. The writing style of the authors tend to be too superfluous and descriptive. Statements should be simple, precise, and get to the point.
Tables 1 and 2 are not “Chemical composition” but “Elemental distribution.”
Author Response
We are grateful for Your suggestions and You can find our answers to Your comments in the attached file.

Reviewer 2 Report
Please find my review attached. I believe that this is a reasonably good paper, which deserves publication after some minor revision. However, it is too long, such that it was a real slog to get through. Also, I feel as if the authors are not terribly familiar with the basics of hydrometallurgy.

Author Response
Dear reviewer,
We are grateful for Your suggestions and You can find our answers to Your comments in the attached file.

Reviewer 3 Report
The study of Jovanovic et al. is a comprehensive investigation of the leaching of Printed Circuit Boards (PCB) with the ultimate scope of recovering metals recovery. The authors tested many parameters and proposed a comprehensive set of experimental data. At the same time the manuscript is excessively long (this is reflected also in a long list of references) and formatted as a long list of data that may not interest all readers. There are also several speculations on the chemistries involved rather than hard data that could support them. I have provided some comments below. Further to those I suggest the authors to be more concise.
- Manuscript
Line 56. I am not sure how hydrometallurgical processing can be less hazardous than pyrometallurgical. Usually hydrometallurgy involves very concentrated acids, solvents, additives of all sorts that are typically hazardous for both the environment and the operators. Later on, in the manuscripts even cyanides are mentioned. In the experiments very high concentrations of peroxides are used which also question the practical aspects of hydrometallurgical routes. Pyrometallurgical processes should be easier and safer.
Line 90. Is glycine abundant? Please provide a reference or elaborate.
Line 105. This is pretty generic. As stated by the authors themselves there are many different types of PCB out there. How was Table 1 determined? Is this a mix of XRD and chemical digestions? If it is just digestions they may have not gone to completion.
Paragraph 2.2. Are the metal concentrations determined via ICP?
Line 235-Line311. I am not sure I understand the logic of this very long paragraph. The authors are just reporting literature findings or speculations. The S/L study does not inform about any of the proposed chemical reactions
Line 404. Was there any direct evidence that HF had formed in the system?
- Language formatting
Line 46. What is it meant by metalloids here? Is this the definition from classic chemistry?
Line 47. I assume these are wt%. It would be good to include some sort of graph that shows the different chemical components in the PCB and possibly also the different metals
Line 103. This is very abrupt. It is also not clear how what is the true contribution of the manuscript from this paragraph
Line 145. Leaching is mispelled
Author Response

(The authors gave the same response as above.)

Round 2
Reviewer 1 Report
The paper describes a recycling process in which various metals are recovered from end-of-life printed circuit board (PCB) by standard leaching process. The system is inherently complex, and the objectives and analyses are somewhat confusing, but the paper does have interesting merits to benefit interested readers.
Author Response
It is true that the system is very complex, and we tried to be as concise as possible to avoid confusion. We also believe that the paper does have strong merits to benefit interested readers. Thank You for the review!
Reviewer 3 Report
The authors have provided some amendments and corrections. However, in my opinion those changes did not result in a significant change in the quality of the manuscript. There seem to be un unbalance between overdetailed paragraphs in some areas and lack of fundamental discussions in others. As additional note, the provided reference on the abundance of glycine is not relevant and needs to be changed.
Author Response
The reference has been changed with new reference from the journal relevant to our area of study, and the term abundance replaced with easily accessible. In addition the imbalance referred to has been addressed. Thank You for the review!